# Time-resolved crystallography of boric acid binding to the active site serine of the β-lactamase CTX-M-14 and subsequent 1,2-diol esterification

Andreas Prester [1,8], Markus Perbandt [2], Marina Galchenkova[3], Dominik Oberthuer [3], Nadine Werner[2,9], Alessandra Henkel [3], Julia Maracke[3], Oleksandr Yefanov[3], Johanna Hakanpää[4], Guillaume Pompidor[4], Jan Meyer[4], Henry Chapman [3,5,6], Martin Aepfelbacher[1], Winfried Hinrichs [7], Holger Rohde [1] ✉ & Christian Betzel [2,5] ✉

The emergence and spread of antibiotic resistance represent a growing threat to public health. Of particular concern is the appearance of β-lactamases, which are capable to hydrolyze and inactivate the most important class of antibiotics, the β-lactams. Effective β-lactamase inhibitors and mechanistic insights into their action are central in overcoming this type of resistance, and in this context boronate-based β-lactamase inhibitors were just recently approved to treat multidrug-resistant bacteria. Using boric acid as a simplified inhibitor model, time-resolved serial crystallography was employed to obtain mechanistic insights into binding to the active site serine of β-lactamase CTX-M-14, identifying a reaction time frame of 80–100 ms. In a next step, the subsequent 1,2-diol boric ester formation with glycerol in the active site was monitored proceeding in a time frame of 100–150 ms. Furthermore, the displacement of the crucial anion in the active site of the β-lactamase was verified as an essential part of the binding mechanism of substrates and inhibitors. In total, 22 datasets of β-lactamase intermediate complexes with high spatial resolution of 1.40–2.04 Å and high temporal resolution range of 50–10,000 ms were obtained, allowing a detailed analysis of the studied processes. Mechanistic details captured here contribute to the understanding of molecular processes and their time frames in enzymatic reactions. Moreover, we could demonstrate that time-resolved crystallography can serve as an additional tool for identifying and investigating enzymatic reactions.

Esterification of boric acid with organic *cis*-diol-functionalized molecules appears to be the most probable chemical mechanism of boron compounds in biological systems, for example, sugar-binding properties and the involvement of boric acid and borates in the evolution of the living world[1–3]. In the past two decades, the status of boronic acid derivatives in biochemistry has gone from that of peculiar and rather neglected compounds to that of a prime class of synthetic compounds in their own right. Impressive advances have been made in the use of boronic acids in molecular recognition, materials science, catalysis and medicine[1,2,4]. Serine proteases in detergents are stabilized as boric acid esters preventing self-digestion[5]. Crystal structures of boronic acid adducts of subtilisin BPN' (Novo) verified the inhibition by specific esterification of the active site serine with boric acid

[1]Institute of Medical Microbiology, Virology and Hygiene, University Medical Center Hamburg-Eppendorf UKE, Hamburg, Germany. [2]Institute of Biochemistry and Molecular Biology, University of Hamburg, Hamburg, Germany. [3]Center for Free-Electron Laser Science CFEL, DESY, Hamburg, Germany. [4]Deutsches Elektronen-Synchrotron DESY, Hamburg, Germany. [5]Hamburg Centre for Ultrafast Imaging CUI, University of Hamburg, Hamburg, Germany. [6]Department of Physics, University of Hamburg, Hamburg, Germany. [7]Institute of Biochemistry, University of Greifswald, Greifswald, Germany. [8]Present address: Institute of Biochemistry and Signal Transduction, University Medical Center Hamburg-Eppendorf UKE, Hamburg, Germany. [9]Present address: Centre for Integrative Biology, Department of Integrated Structural Biology, Institute of Genetics, Molecular and Cellular Biology, IGBMC, Illkirch, France. ✉e-mail: rohde@uke.de; christian.betzel@uni-hamburg.de

and derivatives[6]. Boric acid reacts as Lewis acid with the active site serine and mimics the tetrahedral transition state of the catalytic mechanism of serine proteases and also of β-lactamases (see Supplementary Fig. 1).

The approval of the first boronic acid-containing drugs like the anticancer agents bortezomib and ixazomib and the β-lactamase inhibitors vaborbactam and taniborbactam further confirm the growing status of boronic acid derivatives as an important class of compounds in chemistry and medicine[7–12]. In particular, the ability to repurpose boric acid derivative inhibitors of other target proteins, such as the proteasome inhibitors bortezomib and ixazomib, to inhibit serine β-lactamases (SBLs) highlights the exceptional potential and importance of boric acid derivatives in medicinal chemistry[13]. The active sites of serine proteases and SBLs share the same key structural features required for the catalytic mechanism. β-Lactamases are enzymes that hydrolyze the most important class of antibiotics, the β-lactams, enabling pathogens to acquire resistance by expression of these proteins. These β-lactamases are divided into four Ambler classes (A–D) according to their sequence[14–16]. Classes A, C and D belong to the SBLs, whereas class B represents the metallo-β-lactamases (MBLs). Boric acid itself is known to act as a weak reversible competitive inhibitor of SBLs[17,18]. An increasing number of boronate-based β-lactamase inhibitors are being investigated recently[10,11,19,20], such as mono- and bicyclic boronates[21]. These boric acid derivatives are developed to increase the specificity and binding affinity to the target enzymes. Furthermore, 3-aminophenylboronic acid is used as a potentiator in disc diffusion sensitivity assays for the detection of *Klebsiella pneumoniae* (*K. pneumoniae*) carbapenemase-type β-lactamase positive isolates. It increases the inhibition zone diameter on the agar plate and allows for differentiation from those isolates producing other types of β-lactamases[22]. As boronate-based compounds are recognized for their ability to inhibit β-lactamases we focused our investigations, towards the time-resolved analysis of the enzymatic reaction and the associated temporal processes of the β-lactamase CTX-M-14 from *K. pneumoniae* in presence of boric acid. Given that antibiotic resistance remains one of the greatest threats to global health, a thorough comprehension of the associated enzymatic mechanisms is immensely valuable[23–25]. Recently, advanced methods and techniques of serial mix-and-diffuse crystallography have been developed applying tape drive[26,27], jet[28,29], or fixed target[30,31] sample delivery techniques. Correspondingly, the growing potential of time-resolved serial crystallography allows to obtain completely new insights into reaction processes and associated structural dynamics at different time points as demonstrated in recent structural enzymology studies[32–34].

Recent publications applying time-resolved crystallography reported the inhibition of an SBL from *Mycobacterium tuberculosis* by sulbactam in millisecond resolution time steps and the bond cleavage of a β-lactam ring by an MBL from *Stenotrophomonas maltophilia*[35,36]. We studied the structural kinetics of boronate-based inhibition by employing boric acid binding to the active site serine of a β-lactamase as a simplified model reaction. Mix-and-diffuse serial crystallography was applied to determine time-resolved distinct structures and structural intermediates of the class A SBL CTX-M-14 from *K. pneumoniae* during boric acid ester formation with the active site serine using the TapeDrive system[26]. Additionally, we demonstrate the time-resolved observation of the subsequent 1,2-diol esterification of the boric acid serine ester with glycerol in the active site of CTX-M-14. Our results provide detailed insight into the stepwise binding of boronic acid and subsequent reactions occurring in the active site of the enzyme with high stereoselectivity.

## Results
### CTX-M-14 microcrystals offer perfect conditions for mix-and-diffuse experiments

The TapeDrive system[26,27,37] was applied to collect serial diffraction data at the beamline P11, PETRA III/DESY, to explore the kinetics and structural intermediates of ligand binding to the β-lactamase CTX-M-14. As a result, protein structures with delay times of 50–10,000 ms and a resolution range of 1.40–2.04 Å were obtained. For this purpose, CTX-M-14 microcrystals were mixed with boric acid to initiate the binding process, and diffraction data were collected after distinct pre-set delay times. To monitor the formation of a diester, microcrystals were pre-soaked with boric acid and subsequently mixed with glycerol and again diffraction data were collected after distinct delay times. The obtained data can reveal the time evolution of populations and, as for all mix-and-diffuse serial crystallography data collections, can represent multiple states in one structure. The delay times, the corresponding PDB entries, the obtained diffraction quality and model refinement statistics are summarized in Supplementary Tables 1 and 2. In our own unpublished experiments, macro-crystals of CTX-M-14 were soaked with boric acid and diffraction data were collected by conventional rotation crystallography at cryo-conditions. Glycerol was used as a cryo-protectant, and thus the cyclic glycerol boric acid diester (GBE) in the active site described here has been observed (PDB code 8r7m). However, a time-resolved analysis of the processes seemed very intriguing due to the two sequential reactions. To observe the reactions via time-resolved crystallography applying the available TapeDrive setup of CFEL at PETRA III, DESY, the reaction rates needed to be decreased. In terms of pilot investigations, we observed that $k_{cat}$ is reduced approximately twofold at pH 4.5 compared to pH 7.4. Therefore, the relatively low pH 4.5 applied for the crystallization conditions supported the optimization of time-resolved diffraction data collection of CTX-M-14, although it does not correspond to the physiological pH value. The asymmetric unit of CTX-M-14 crystals contains one monomer with the active site region solvent accessible. The Matthews coefficient of the crystals is 2.15 Å³/Da, corresponding to a solvent content of 43%. The solvent channels in the crystal lattice allow rapid diffusion of low molecular weight ligands to the active site, scoring the CTX-M-14 crystals to be ideal for time-resolved serial crystallographic investigations, applying the TapeDrive mixing approach[27]. Furthermore, the small but excellent diffracting crystals of CTX-M-14 with dimensions of 11–15 μm have a relatively small ligand distribution period within the crystal lattice and due to short diffusion times exhibit sharper delay time points compared to larger crystals[38]. As a reference, the TapeDrive was also used to collect serial data of the native CTX-M-14 crystals. The occupancies of boric acid and its glycerol diester in the active site obtained after different mixing time points were refined and compared to discuss the stepwise rearrangements in the active site in detail below. To avoid correlation of occupancies and B-factors during refinement, care was taken that between datasets of adjacent time points the individual B-factors of the respective ligands did not differ more than the Wilson B-factors and the average B-factors (Supplementary Fig. 2).

### Boric acid binding interferes with the anion-binding site of CTX-M-14

CTX-M-14 has a crucial anion-binding site (Fig. 1) close to the active site residues that is occupied by the carboxylate of β-lactam substrates[39]. In the native enzyme, this site is occupied by a tetrahedral anion, such as a phosphate (PDB code 4ua6[40]) or a sulfate (PDB code 7q0z[13]), as in the structures we refined (Figs. 1 and 2). In a structure of CTX-M-14 in complex with ixazomib/bortezomib (PDB code 7q11/7q0y[13]), the inhibitor does not directly occupy the anion-binding site, but still displaces the bulky tetrahedral anion, which is replaced by a smaller chloride to balance the charge[13].

The rotationally disordered sulfate occupies two slightly displaced alternative positions in the native enzyme (SO4-A and SO4-B, S to S distance of 0.4 Å, Fig. 1). The alternative sulfates (A/B) are coordinated via hydrogen bonds with the side chains of Ser70 (3.1 Å/3.4 Å), Thr235 (3.1 Å/3.3 Å) and Ser237 (3.4 Å/2.8 Å) as well as the main chain nitrogen of Ser237 (3.1 Å/3.2 Å) (Fig. 2a). SO4-B forms additional hydrogen bonds with side chains of Ser130 (2.9 Å) and Lys234 (3.2 Å) (Fig. 2a). During boric acid binding, the sulfate is reoriented in such a way that it is more distant to the Ser70 side chain and coordinated by hydrogen bonds with the side chain hydroxyl groups of Ser130 (3.1 Å), Thr235 (3.0 Å) and Ser237 (2.9 Å) (Figs. 2b and 3). In addition, the boric acid O2 (2.6 Å) can act as a hydrogen bond donor for the sulfate. The esterification with glycerol finally displaces the sulfate ion, since the equivalent O2 of the cyclic diester cannot act as a hydrogen bond donor anymore and due to steric competition (Fig. 2c). In

**Fig. 1 | Representation of the CTX-M-14 active site in the native state, the state with bound boric acid and the state with subsequently formed glycerol boric acid diester. a** Cartoon plot of the CTX-M-14 β-lactamase from *Klebsiella pneumoniae* and close-up views of the active site in surface representation of **b** the native constitution with a sulfate ion (SO4-A magenta; SO4-B yellow) in the anion-binding site, **c** with bound boric acid (BAB, pink) and a sulfate ion, and **d** with bound glycerol boric acid diester (GBE, pale green). BAB (**c**) and GBE (**d**) complex structures are shown with mixing delay times of 10 s, respectively.

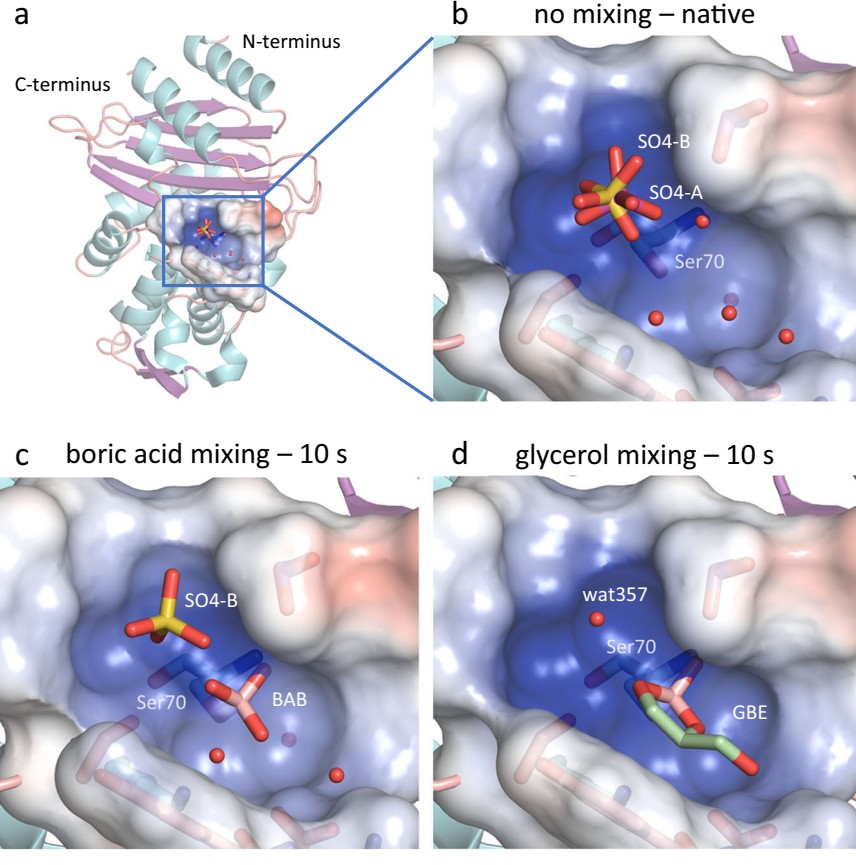

the electron density maps substantially reduced density is observed at this site (Fig. 4). After complete formation of the cyclic diester with boric acid and glycerol, a water molecule (OW357) occupies the position of the anion-binding site. Unlike the sulfate ion, the OW357 can act as a hydrogen bond donor and forms a hydrogen bond with O2 of GBE (2.7 Å). The water molecule OW357 is further stabilized by a hydrogen bond with the hydroxyl group of Thr235 (2.8 Å), as well as a weak hydrogen bond with the main chain carbonyl of Thr235 (3.5 Å).

A direct comparison with recently approved inhibitors such as relebactam (PDB code 6qw8[41]) and avibactam (PDB code 6gth[42]) also shows that utilization of the anion-binding site supports the complex formation. These complexes are stabilized by hydrogen bonds and consequently, the affinity and overall activity of these inhibitors are increased. Accordingly, the sulfonate groups of these new diazabicyclooctane inhibitors occupy the anion-binding site discussed here (Supplementary Fig. 3)[13]. In addition, vaborbactam (PDB code 6v7h[43]) and taniborbactam (PDB code 6sp6[12]) are bound and coordinated in the active site in a similar way. The carboxylate appendage of their oxaborine or benzooxaborine moieties also occupies the anion-binding site[10,12,43]. Thus, for inhibition of SBLs, it is evident that the anion-binding site of the native enzyme is occupied by the inhibitor, supporting enhanced binding if inhibitors feature a suitable moiety that can bind in this region (Supplementary Fig. 3). This anion-binding site represents a very important structural feature of β-lactamases, to be considered in future drug development investigations. In this context, our data are unique, as we show via time-resolved crystallography the time course of the displacement of a sulfate ion from this particularly important binding site.

In addition, the oxyanion hole is utilized by a number of inhibitors forming hydrogen bonds with Ser70 NH and Ser237 NH (see Supplementary Fig. 4). Furthermore, these structural features are also used in the binding modes of β-lactam substrates such as ceftazidime (Supplementary Fig. 4h), as well as in multiple other β-lactamases.

## Boric acid binds to the active site Ser70 within 80–100 ms

The obtained refined time-resolved crystal structures provided insight into the molecular kinetics of the binding of boric acid (Fig. 3 and Supplementary Fig. 5). Starting from the native CTX-M-14 structure, the above-mentioned sulfate and some water molecules (notably OW174, OW352, OW353 and the catalytic OW10) are present in the active site well-defined in the electron density maps. At a delay time of 50 ms after mixing the microcrystals with boric acid initially, no additional electron density for the boric acid was observed. Meanwhile, the electron density of the sulfate ion has already changed, indicating a slight shift between the two alternative locations. Initially, in the native enzyme, the alternative positioned sulfate ions refined to occupancies of 47% and 44% for SO4-A and SO4-B, respectively. These change in the 50 ms structure to occupancies of 54% for the SO4-A and 41% for the SO4-B position also indicate that initially the position closer to the Ser70 is preferred before the boric acid will covalently bind to Ser70 OG. After a delay time of 80 ms, a weak electron density for the bound boric acid (BAB) was observed in the calculated polder map, with a corresponding occupancy of 35%. At the same time, the sulfate ion in the SO4-B position was reoriented by slight translation and rotation so that an oxygen atom has a distance of 2.6 Å to the O2 hydroxyl group of BAB (Fig. 3). The evaluation of the electron density maps revealed that the hydroxyl groups of BAB occupy approximately the same positions as previously occupied by an oxygen of SO4-A and the two water molecules OW352 and OW353. The calculated occupancy for BAB (Fig. 5a, Supplementary Table 3) and the corresponding electron density increased with longer delay times after mixing, resulting in a well-defined electron density for BAB in the calculated polder map after only 250 ms delay time. At this delay time, the occupancy of BAB is already 49%, whereas the occupancy of SO4-A has dropped to 33%. In the further time course investigated, the occupancy of BAB increases only slightly. After a delay time of 10 s, it reaches the maximum occupancy of 53%. Even soaking the CTX-M-14 microcrystals in boric acid for 1 h

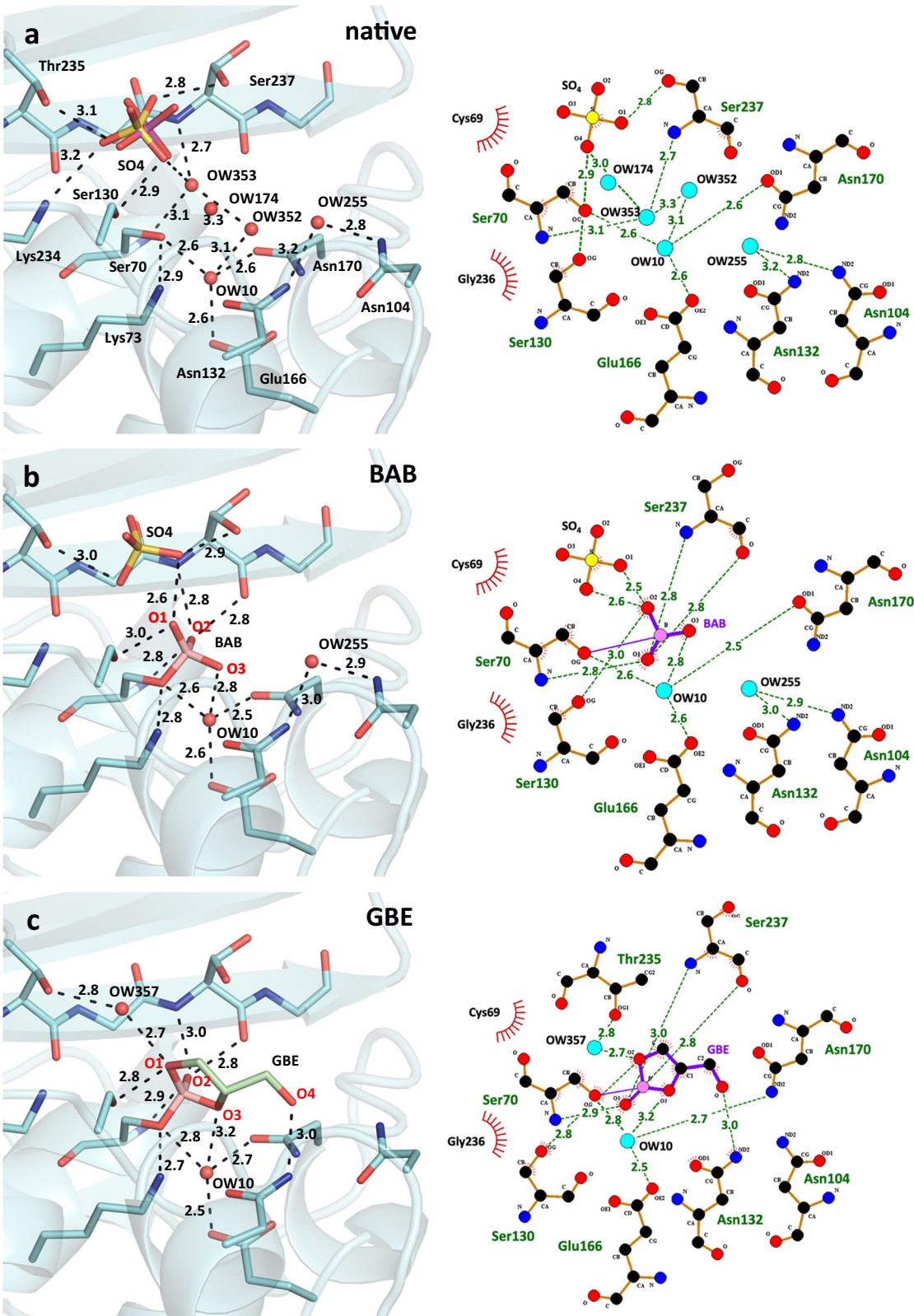

**Fig. 2 | Hydrogen bonding patterns of the individual equilibrium states in the active site of CTX-M-14 in the native, BAB and GBE state.** Stick and cartoon representation (left) as well as 2D-LigPlot+ representation (right) of the active-site amino acid residues highlighting the hydrogen bond network in the native form (**a**), with bound boric acid (10 s, BAB) to Ser70 (**b**) and with the bound glycerol boric acid diester (10 s, GBE) (**c**). Each equilibrium state is displayed individually without overlapping with the initial states. BAB and GBE oxygen atoms are labeled in red. Potential hydrogen bond distances (Å) are indicated by dashed lines.

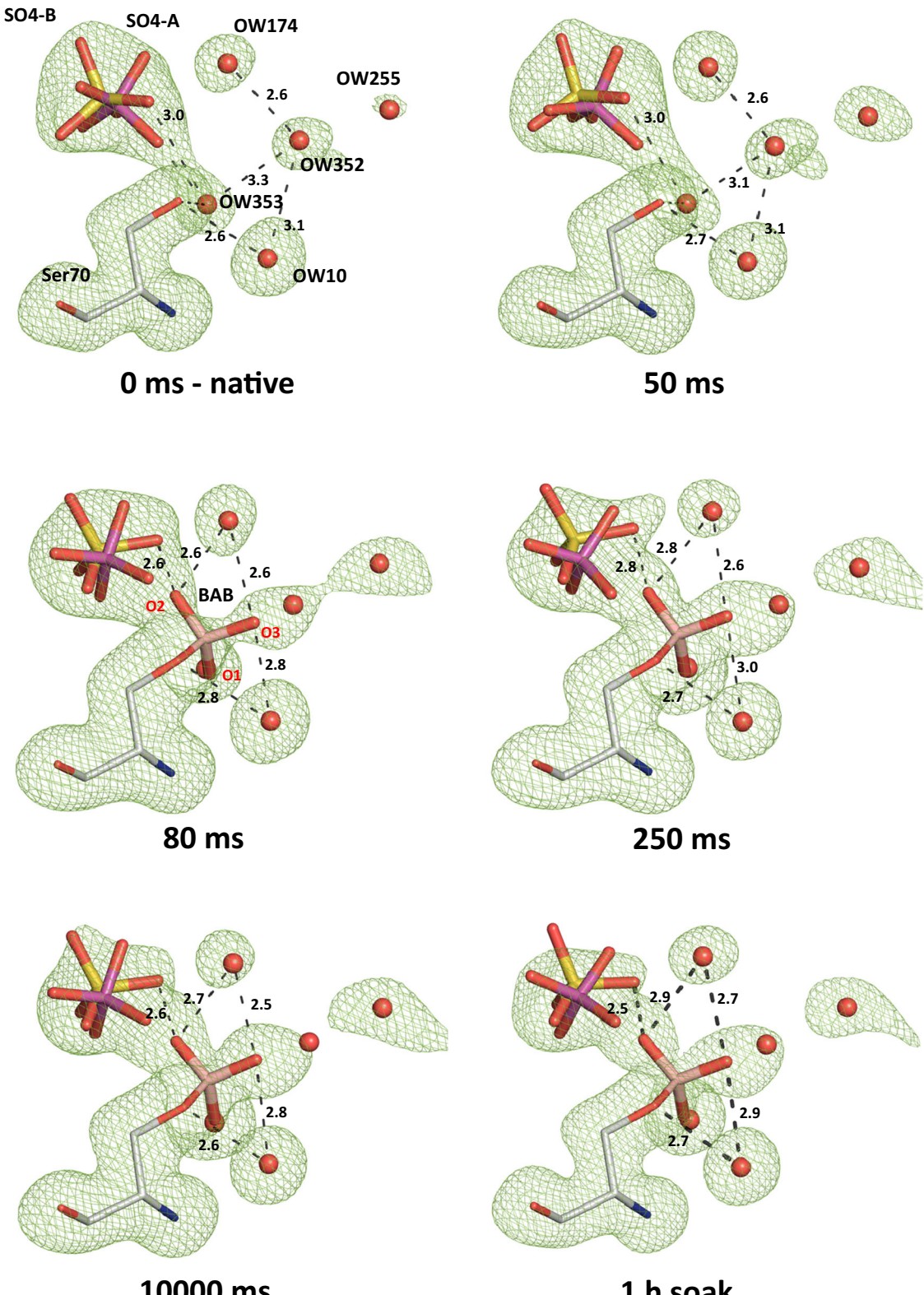

**Fig. 3 | Time-resolved structures for the analysis of the binding process of boric acid to CTX-M-14.** Polder electron density (contoured at 5 σ, green mesh) of the active site Ser70, the sulfate ions and bound boric acid are shown at different delay time points after mixing microcrystals with boric acid. The 1 h soak structure was obtained with the TapeDrive after microcrystals have been soaked in boric acid for 1 h and shows that almost no further increase in electron density is observed after 10 s. BAB and GBE oxygen atoms are labeled in red. Potential hydrogen bond distances (Å) are indicated by dashed lines.

**Fig. 4 | Time-resolved structures for the subsequent esterification of bound boric acid with glycerol in the active site of CTX-M-14.** Polder electron densities (contoured at 5 σ, green mesh) of the active site region of CTX-M-14. Microcrystals pre-soaked with boric acid and mixed with glycerol prior to serial diffraction data collection applying the TapeDrive setup at beamline P11, PETRA III/DESY, observing time-resolved the ester bond formation between glycerol and the Ser70 borate ester. The sulfate anion present in the native conformation is displaced upon binding of GBE and finally replaced by solvent water OW357. BAB and GBE oxygen atoms are labeled in red. Potential hydrogen bond distances (Å) are indicated by dashed lines.

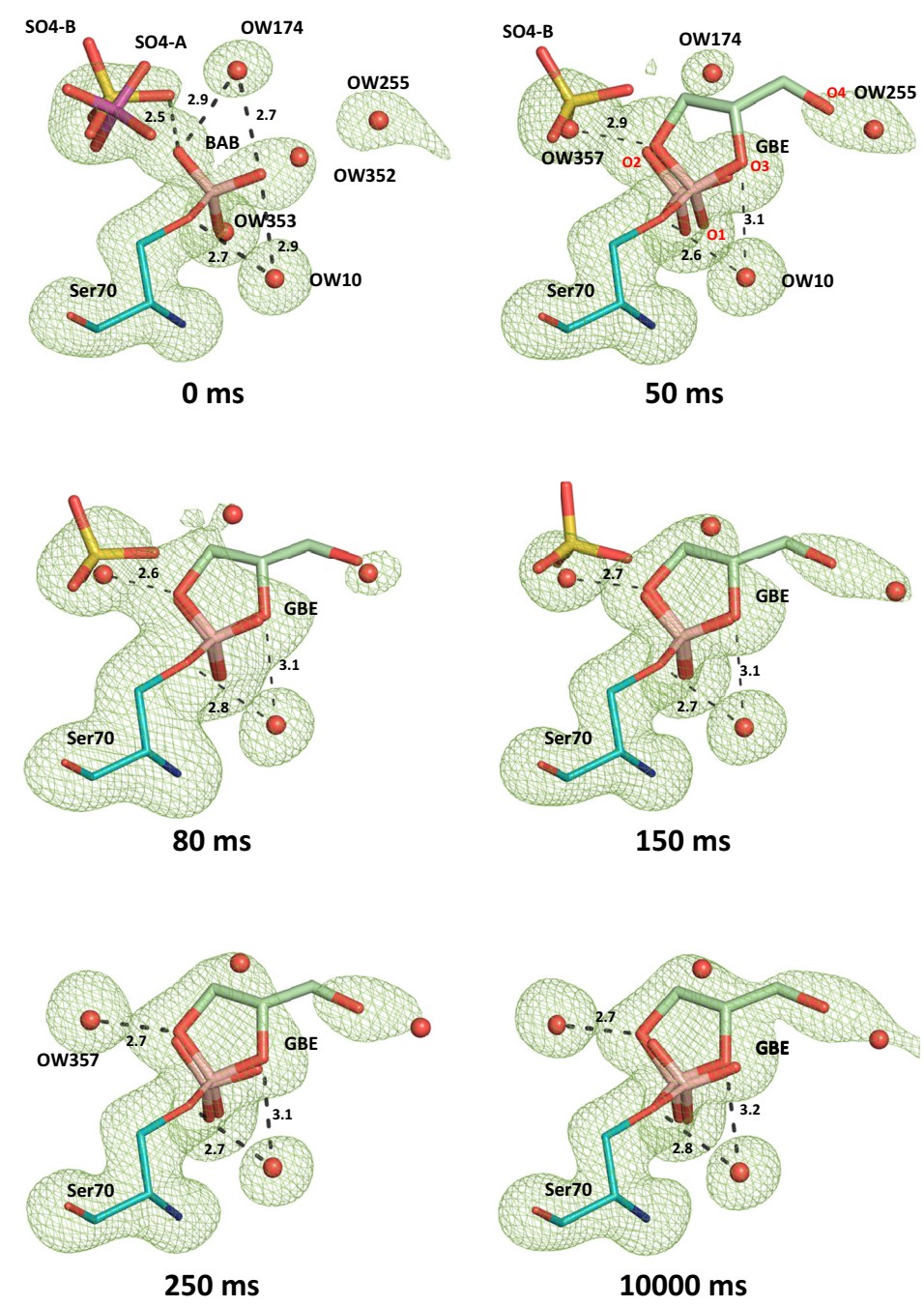

could only increase the occupancy to 57%. This indicates that under these conditions the equilibrium of the BAB formation has been reached.

**Binding mode of boric acid**

Boric acid binds to the active site of CTX-M-14 (Fig. 1c) forming an ester with the Ser70 OG. The hydrogen bonding interactions that stabilize the tetrahedral transition state analog during initial binding include the oxyanion hole (Ser70 NH and Ser237 NH). Similar to the binding mechanism of substrates, the nucleophilic attack of Ser70 OG can be supported via activation of the OG by the general base Lys73[44]. The unprotonated Lys73 side chain can assist in the nucleophilic attack by acting as a general base thereby accepting the proton from the Ser70 OG when the tetrahedral intermediate is formed. A corresponding proposed reaction pathway is shown in Fig. 6. Similar to the carboxylate of the acyl–enzyme intermediate, one hydroxyl group of boric acid (O1) forms hydrogen bonds with the main chain

nitrogen atoms of Ser70 (2.8 Å) and Ser237 (2.8 Å), constituting the oxyanion hole (Fig. 2b). In contrast to bortezomib and ixazomib, the remaining two hydroxyl groups of BAB do not form hydrogen bonds with Asn170 and Glu166[13] (Supplementary Fig. 4). In fact, the boric acid is shifted rather in the opposite direction in the anion-binding site, forcing a reorientation of the sulfate ion from the position of SO4-A to the position of SO4-B (Fig. 2b), to prevent too close atomic contacts. The boric acid is further stabilized in this position via hydrogen bond interactions of the BAB hydroxyl group (O2) with the hydroxyl group of Ser130 (3.0 Å) and the sulfate ion (SO4-B, 2.6 Å). The third BAB hydroxyl group (O3) forms a hydrogen bond with the water molecule OW10 (2.8 Å). In all observed time steps OW10 remains well-defined in the same position. This water molecule is well-known as the catalytic water molecule mandatory for the deacylating step in β-lactam hydrolysis[45], initiated by nucleophilic attack on the carbonyl carbon atom of the acyl–enzyme complex to hydrolyze the acyl bond. It forms hydrogen

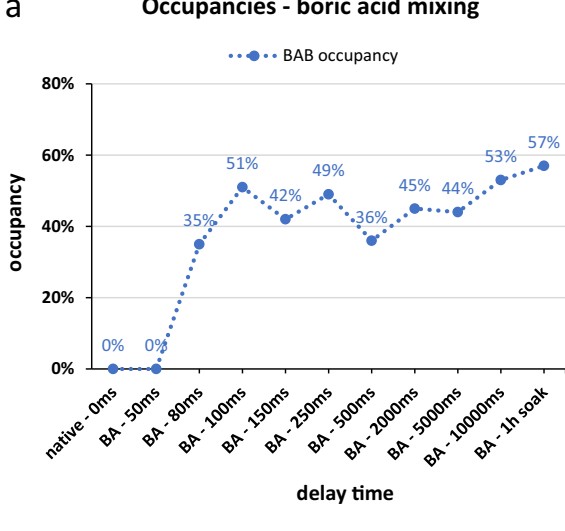

### a  Occupancies - boric acid mixing

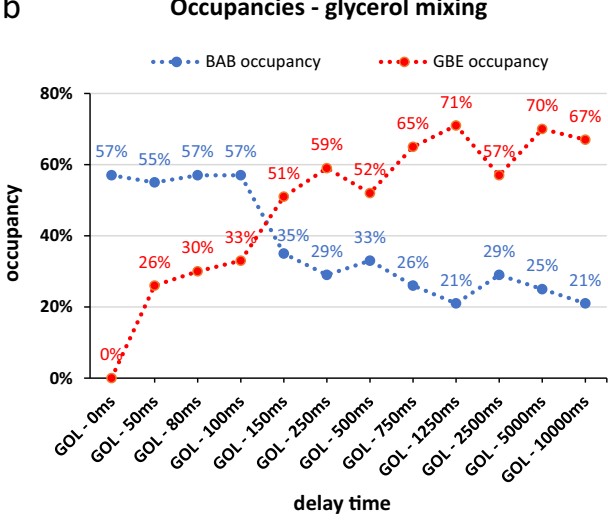

### b  Occupancies - glycerol mixing

**Fig. 5 | Progression of the calculated occupancies of bound boric acid and glycerol boric acid diester during the observed time frame in time-resolved crystallography experiments.** Plots of BAB (**a**) and GBE (**b**) with the refined occupancy values obtained in the context of the respective delay times (no linear display), after mixing with boric acid (BA) or glycerol (GOL). The occupancy of BAB increases with prolonged delay time after mixing with boric acid. Subsequent mixing with glycerol causes the BAB occupancy to decrease again, as it is esterified to GBE. The total boron content continues to increase along mixing with glycerol.

bonds with the side chains of Ser70 (2.6 Å), Glu166 (2.6 Å), Asn170 (2.5 Å) and BAB (O3, 2.8 Å) (Fig. 7). All these intermolecular interactions ensure that BAB is very well coordinated, e.g. a rotational disorder around the Ser70 borate ester linkage is not observed.

### Esterification with glycerol of the boric acid in the active site within 100–150 ms

After monitoring time-resolved structure and dynamics of boric acid binding in the active site of CTX-M-14, we have further investigated the esterification process of boric acid with glycerol. For this purpose, the TapeDrive setup was used again to mix glycerol with CTX-M-14 microcrystals complexed with boric acid beforehand. We defined the delay time 0 ms as the starting condition where no glycerol was added, corresponding to the last time point (1 h soak) of the serial data collection with boric acid, considering that CTX-M-14 microcrystals were saturated with boric acid (Fig. 4 and Supplementary Fig. 6). At this defined time point, the occupancy of BAB was refined to 57%. The first change in the electron density of the

**Fig. 6 | Proposed reaction pathway of the binding process of boric acid (blue) to the active site serine of CTX-M-14.** Hydrogen bonds are displayed as dashed lines.

polder map appears already at the 50 ms mixing/delay point. In the region of the BAB hydroxyl groups extending electron density was observed indicating the formation of a glycerol diester. The obtained electron densities allowed the insertion and refinement of a glycerol boric acid diester (GBE), resulting in a GBE occupancy of 26%, while the BAB occupancy remained almost the same with 55%. This indicated also that the formation of the GBE increases the total occupancy of bound ligand in the active site to 81%. The electron density of the sulfate decreased for SO4-A to zero, as the newly formed glycerol diester occupies this position. The alternatively positioned SO4-B fits into the active site together with the BAB and is therefore still present with the same occupancy as the BAB. The observed electron densities at the 80 and 100 ms delay times showed only a slight increase for GBE occupancy. A sharp increase in the corresponding GBE occupancy to 51% was observed and refined at the 150 ms time point, while in parallel the BAB occupancy dropped to 35% (Fig. 5b, Supplementary Table 3). By this time, all atoms of GBE are covered with the calculated polder electron density. At the 750 ms time point, the entire GBE was well-fitted and covered in the calculated electron density map with a resulting occupancy of 65%. Consequently, since the GBE can no longer act as a hydrogen bond donor for SO4-B due to the lack of hydrogen atoms at the position O2, the sulfate ion is finally completely replaced by a water molecule, OW357, which is accompanied by an increasing GBE and a decreasing BAB occupancy. This correlates with reduced electron density in the SO4 site. The O3 of GBE can also no longer interact with OW10 as a hydrogen bond donor, but only as a hydrogen bond acceptor. GBE approached a refined occupancy of 67% after only 10 s delay time, while BAB occupancy dropped to 21%. However, it is interesting to note that the overall occupancy of the ligands (BAB, GBE) bound to Ser70 increased with the observed increase in electron density obtained and refined for the cyclic diester. Thus, the total occupancy of the binding site and region increased from 57%, obtained for soaking only with boric acid, up to 88% when further mixing with glycerol up to a delay time of 10 s. The stepwise blocking of the active site by boric acid and the subsequent glycerol diester formation is shown in Fig. 4.

### Binding mode of the glycerol boric acid diester

Boric and boronic acids have a propensity to form esters with polyalcohols, resulting in the formation of five- or six-membered rings[46–48]. The observed five-membered scaffold of GBE is reminiscent of the autoinducer-2. This borate diester was first observed in complex with the sensor protein LuxP of the marine bioluminescent bacterium *Vibrio harveyi*[49]. The triol glycerol can alternatively form both ring systems, with the formation of a six-membered ring being energetically preferred over the five-membered ring, as shown in a computational study[46]. The investigation of peptidomimetic-boronic acid inhibitors for flaviviral proteases revealed both, a five-membered ring formation of the boric acid moiety and glycerol in the active site for the West-Nile virus NS2B–NS3 protease and a six-membered ring formation for the Zika virus NS2B–NS3 protease[47,48]. Despite the high similarity of these

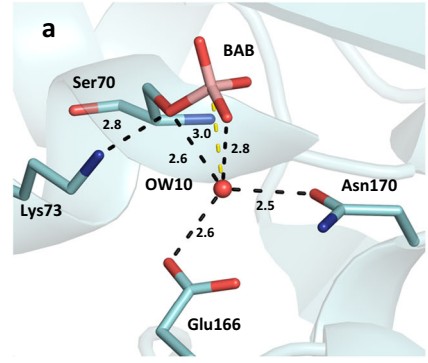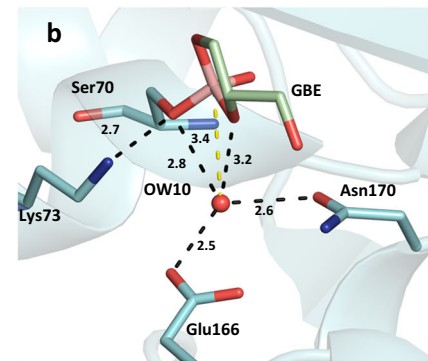

**Fig. 7 | Hydrogen bonding pattern of the catalytic water molecule OW10 in the active site of CTX-M-14 with bound boric acid and bound glycerol boric acid diester.** The active site of the (**a**) bound boric acid and (**b**) glycerol boric acid diester is shown at the 10 s delay time point. OW10 is hydrogen bond donor and acceptor to the boric ester of Ser70 (2.6 Å/2.8 Å). The tetrahedral hydrogen bonding pattern of OW10 is completed by Glu166 (2.6 Å) and Asn170 (2.5 Å). Hydrogen bonds of OW10 to GBE are longer than to BAB (2.8 Å/3.2 Å) while the hydrogen bonding pattern with Glu166 (2.5 Å) and Asn170 (2.6 Å) remains similar. The boron atom is positioned at a distance of 3.0 Å (BAB, 10 s) or 3.4 Å (GBE, 10 s) from the catalytic water OW10. Thus, the catalytic water could perform a nucleophilic attack on the boron atom, leading to the reversible hydrolysis of the boric acid serine ester linkage in BAB and GBE. Potential hydrogen bond distances (Å) are indicated by dashed lines.

**Fig. 8 | Proposed reaction pathway of the esterification of glycerol (green) with boric acid bound to CTX-M-14.** Hydrogen bonds are displayed as dashed lines.

enzymes, both ring formations were observed, clearly showing the influence of the individual active site, resulting in a preference due to steric constraints[47,48]. In the CTX-M-14 active site, glycerol forms a five-membered cyclic diester with two of the three hydroxyl groups (O2, O3) of boric acid that is bound to the active site Ser70 (Fig. 2c). A corresponding proposed reaction pathway is shown in Fig. 8. The remaining hydroxyl group (O1) of the boric acid maintains the stabilizing hydrogen bonds with the main chain nitrogen atoms of Ser70 (2.9 Å) and Ser237 (3.0 Å) in the oxyanion hole (Fig. 2c). During the esterification the sulfate ion in the anion-binding site is finally replaced by a water molecule (OW357) that forms alternative hydrogen bonds with the cyclic diester O1 (2.7 Å) and the hydroxyl group of Thr235 (2.8 Å) (Fig. 2c). The other oxygen of the cyclic diester O3 forms a hydrogen bond with OW10 (3.2 Å), which itself is strongly coordinated by Ser70 (2.8 Å), Glu166 (2.5 Å) and Asn170 (2.7 Å). The remaining free hydroxyl group of GBE (O4) forms an additional hydrogen bond with the amide side chain of Asn132 (3.0 Å) and weak hydrogen bonds with amide side chains of Asn104 (3.5 Å) and Asn170 (3.5 Å) (Fig. 2c). In that conformation all oxygen atoms of the GBE are coordinated via hydrogen bonds either directly with the enzyme or via a water molecule. This is probably also the reason for the preference of the five-membered over the six-membered cyclic diester in the CTX-M-14 active site. In a six-membered ring, the free hydroxyl group could not form hydrogen bonds with Asn132 because it would be located in the center of the molecule. In fact, there would probably be no side chain for possible

hydrogen bond interactions with the free hydroxyl group in that orientation as it would point out of the active site. Thus, the formation of a hydrogen bond of the free hydroxyl group of GBE with Asn132 is probably the determining factor, explaining our observation of only five-membered cyclic diester formation in all obtained GBE structures.

The central carbon atom of the glycerol diester with boric acid becomes a stereo center with *S*-configuration. Also, the boron atom of GBE is a stereo center with *S*-configuration. Both stereocenters are observed without any racemic disorder. This is probably an indication for the specific active site environment of the β-lactamase. For example, proteinase K has weak specific substrate preferences and glycerol forms a simple monoester with the boric acid bound to the active site serine (PDB code 2id8[50]). Obviously, the stepwise formation of a monoester and diester is much too fast to be observed with our experimental setup.

### Inhibition of β-lactamase activity by boric acid and a combination of boric acid and glycerol

As expected, the covalent binding of boric acid and the boric acid diester to the catalytic Ser70 in the active site of CTX-M-14 β-lactamase resulted also in the inhibition of the enzyme[17,18]. Boric acid remains in the active site of the β-lactamase in the crystal lattice with an occupancy of 57% even after prolonged soaking. Consequently, it can be concluded that the boric acid diester does not dissociate over time and therefore inhibits the enzyme (in the crystal lattice) for a certain period if the solvent conditions are

unchanged. To quantify the effect of the observed occupation of the active site, enzymatic activity assays applying a photometric determination of the 50% inhibitory concentration ($IC_{50}$) values were performed. Moderate $IC_{50}$ values of 2.9 ± 0.4 mM for boric acid and 3.1 ± 0.4 mM for the combination of boric acid with glycerol were determined (Supplementary Fig. 7). Interestingly, the $IC_{50}$ values are quite similar even though the crystallographic data showed a higher occupancy of the GBE in the crystal lattice, which would imply a higher inhibition. Compounds that are considered as inhibitors usually have substantially lower $IC_{50}$ values, therefore the boric acid and the glycerol diester at this point cannot be considered as effective β-lactamase inhibitors. This is in line with the observed incomplete occupancy of the boric acid and its glycerol diester in the crystal structures and the potentially reversible binding of boric acid. The organization of the active site in the endpoint complexes may also indicate that reversible mechanism for the dissociation of the inhibitor. The boron atom is positioned at a distance of 3.0 Å (BAB, 10 s) or 3.4 Å (GBE, 10 s) from the catalytic water OW10 (Fig. 7). Thus, the catalytic water is well positioned to perform a nucleophilic attack on the boron atom, leading to the reversible release of boric acid or the GBE. Reversible inhibitors have the advantage of not being depleted or modified by their target, thereby enabling their capacity to inhibit several enzymes during their lifetime. Our data highlight the potential of boric acid derivatives in medicinal chemistry.

## Conclusion

Applying serial crystallography we characterized time-resolved structures and kinetics of boric acid binding in the active site of CTX-M-14 in a resolution range of 1.4–2.0 Å. Particularly we could monitor structure and structural changes in the time period of 50–100 ms, considering and emphasizing also the presence of a functional anion coordinated in the active site region. We showed that boric acid plays a versatile role upon inhibiting a β-lactamase, which also explains and highlights why boric acid-based compounds are currently in the focus of recent drug discovery investigations[10,11,13,19,21]. Moreover, to the best of our knowledge we showed for the first time via high-resolution structures that boric acid bound to active site Ser70 of CTX-M-14 is capable to perform chemical reactions, such as the esterification with glycerol. This diester formation takes place within a time frame of 100–150 ms after adding glycerol to microcrystals of boric acid inhibited β-lactamase. Such two consecutive reactions have not been shown in an active site of a β-lactamase before. However, the stepwise formation of monoester and diester was not observed, suggesting their reaction rates are far too rapid for the applied experimental setup to be captured. The formation of the monoester brings the other glycerol hydroxyl groups and the boric acid into close proximity, probably accelerating the formation of the second ester bond to form the cyclic diester. With the formation of the cyclic diester, two stereocenters were formed, which were observed without any racemic disorder, clearly highlighting the stereoselectivity in the active site. This is further highlighted by the formation only of a five-membered cyclic diester of boric acid and glycerol. In this case, no electron density was observed for a possible six-membered ring, as was the case in the NS2B–NS3 proteases, for example[47,48]. Nevertheless, the formation of the cyclic diester shifted the equilibrium toward an occupied active site state. This was confirmed by the calculated total occupancies of the ligands, which increased from 57% for BAB alone to ~88% for BAB/GBE.

A limitation of the study is the use of a simplified inhibitor model, as the observed binding processes can only be transferred to real inhibitors to a limited extent. Time-resolved X-ray crystallography with much larger ligand molecules is more difficult due to the longer diffusion times into the crystal and also the higher number of possible conformations in the active site. Ultimately, X-ray crystallography is an ensemble method of many individual molecules in a crystal and cannot provide a clear depiction of heterogeneous samples. An exact assignment of the electron density would be difficult in such a case. Nevertheless, the information about the reaction time frames and also the observed processes, such as the displacement of the sulfate, are important observations that enhance our understanding about

molecular processes during ligand binding in the active site of this enzyme. Our investigations also demonstrate the feasibility and potential of time-resolved crystallography studies in enzymology. Moreover, we demonstrate that time-resolved crystallography can serve as an additional tool for identifying and studying enzymatic reactions.

## Methods

### Sample preparation

β-lactamase CTX-M-14 from *K. pneumoniae* was produced, purified and crystallized as described before[42], with a slight modification of the crystallization conditions to obtain crystals matching approximately the X-ray focal spot of 4 × 8 μm. In short, pCR4:CTX-M-14 plasmid was transformed in competent *Escherichia coli* BL21 (DE3) using heat shock. The cells were grown at 37 °C in LB medium with 100 μg ml$^{-1}$ ampicillin for plasmid selection. Protein expression was induced at an optical density of 0.5–0.7 by adding isopropyl β-D-1-thiogalactopyranoside (IPTG) to a final concentration of 150 μM. Cells were harvested after 3 h by centrifugation with 5500 × *g* at 4 °C for 10 min. The cells were lysed by sonication and the supernatant was then dialyzed over night against 20 mM MES, pH 6 using a 10 kDa molecular weight cut-off membrane. The protein was purified using cation exchange chromatography (5 ml HiTrap SP FF, Cytiva) and eluted using a gradient of 20 mM MES, pH 6, 0–50 mM NaCl over 5 column volumes. The protein was concentrated to 22 mg ml$^{-1}$ using 10 kDa Amicon Ultra-15 centrifugal filter units. Applying the batch crystallization procedure, 50% (v/v) CTX-M-14 solution (22 mg ml$^{-1}$) was mixed with 45% (v/v) crystallizing agent [40% (m/v) PEG8000, 200 mM lithium sulfate, 100 mM sodium acetate, pH 4.5] and with 5% (v/v) undiluted seed stock solution. Crystals with a homogeneous size distribution of 11–15 μm were obtained after ~90 min. Crystals were centrifuged at 200 × *g* for 5 min and the supernatant was replaced with a stabilization buffer [28% (m/v) PEG8000, 140 mM lithium sulfate, 70 mM sodium acetate, 6 mM MES, 15 mM sodium chloride, pH 4.5] to stop further crystal growth. In addition, this ensured that no protein remained in the solution that could react with the added ligands. Prior to serial data collection, the microcrystal suspension and ligand solutions were filtered using a 30 μm Celltrics gravity flow filter (Sysmex Corp).

### Setup at beamline and data collection

A conveyor belt device, called TapeDrive[26], was utilized at the beamline P11 (Petra III, DESY, Hamburg, Germany) for supply of crystal sample suspensions, as previously described[27]. Injection nozzles specifically designed for the TapeDrive were used for sample delivery, which allowed the microcrystal suspension and ligand solution to be delivered separately so that they were mixed together just on the tape, reducing the fastest delay time to 50 ms[37]. Serial diffraction data were collected at room temperature from randomly oriented microcrystals that were mixed with the ligand solution directly on the tape. For the boric acid binding experiments, the sample flow rate of native *K. pneumoniae* β-lactamase CTX-M-14 microcrystals and the flow rate of the boric acid solution [200 mM boric acid, 20% (m/v) PEG8000, 100 mM lithium sulfate, 50 mM sodium acetate, 10 mM MES, 25 mM sodium chloride, pH 4.5] were aligned to 2 μl/min to maintain a 1:1 (v/v) mixing ratio. For the experiments to monitor the esterification with glycerol, the CTX-M-14 microcrystal suspension was first soaked in a boric acid solution [100 mM boric acid, 28% (m/v) PEG8000, 140 mM lithium sulfate, 70 mM sodium acetate, 6 mM MES, 15 mM sodium chloride, pH 4.5] for 1 h and then mixed 1:1 (v/v) with a glycerol solution [20% (m/v) glycerol, 20% (m/v) PEG8000, 100 mM lithium sulfate, 50 mM sodium acetate, 10 mM MES, 25 mM sodium chloride, pH 4.5] directly on the tape. Different delay times for the mixing experiments were achieved by adjusting the tape speed and the distance between the nozzle providing the sample suspensions and the X-ray interaction region accordingly.

### Diffraction data processing

Data collection was carried out using Eiger 2 16M detector. The data were processed using CrystFEL[51,52] (v0.9.1). The peakfinder8 algorithm was used

for identifying the Bragg peaks with parameters: --min-snr = 6 --threshold = 10 --min-pix-count = 2. Detected "hits" were indexed using Xgandalf[53] and using –muti option and integrated with –int-radius = 3,6,8. Dataset of CTX-M-14 was further processed with ambigator[54] in CrystFEL to resolve the indexing ambiguity (from 6/mmm to −3m1_H). This was required as CTX-M-14 crystals have a merohedral space group (P 32 2 1) and exhibit indexing ambiguities. Scaling and merging of the data into point group −3m1_H was carried out applying partialator in CrystFEL, using three iterations and --push-res = 1.0 and figures of merit were calculated using compare_hkl and check_hkl, all part of the CrystFEL package. MTZ files for crystallographic data processing were generated from CrystFEL merged reflection data files using F2MTZ of the CCP4[55] program suite.

## Model refinement

The structure of *K. pneumoniae* CTX-M-14, previously determined from diffraction data obtained at the European XFEL (PDB code 6gth[42]), served as the initial model (after removal of the ligand avibactam). Due to non-isomorphism of the collected datasets with that of 6gth, $R_{free}$ flags were generated randomly using phenix.refine[56,57], and the same set of $R_{free}$ flags was then used for all datasets. Ions and ordered solvent molecules were placed in corresponding electron densities using the program Coot[58]. The ligands were built into the $F_O$–$F_C$ electron density maps and validated by polder omit maps[59]. Due to the overlapping reaction states, the ligands and water molecules of the respective states were grouped (A: SO4-A, OW352 and 353; B: BAB, SO4-B, OW174, 175, 254 and 255; C: GBE and OW357) and their occupancies were refined accordingly to these constrained groups by phenix.refine. In addition, these groups were also labeled as different alternative atom positions (altloc) to prevent unwanted repulsion during refinement. Iterative cycles of restrained maximum likelihood and TLS refinement using phenix.refine and manual model rebuilding using Coot[58] were carried out until convergence. Polygon[60], MolProbity[61], and thorough manual inspection of the obtained models were performed to validate the final models. Data collection and structure refinement statistics are summarized in Supplementary Tables 1 and 2. Figures were generated using PyMOL (The PyMOL Molecular Graphics System, Version 2.0 Schrödinger, LLC) and LigPlot$^+$ [62].

## Inhibition assays

The 50% inhibitory concentration ($IC_{50}$) of CTX-M-14 was determined as the concentration of the inhibitor required to reduce the initial rate of hydrolysis of the substrate by 50%. The compounds were diluted at concentrations ranging from $10^{-5}$–$10^6$ μM in PBS buffer at pH 7.4. Diluted enzyme with a final concentration of 10 nM was added to the inhibitor and the solution was incubated at 37 °C for 15 min. When glycerol was included in the assay, it was added after 10 min incubation with boric acid. Following the incubation, the substrate cefotaxime was added at a final concentration of 100 μM and the hydrolysis of the substrate was monitored in 96-well UV-Star® Microplates (Greiner Bio-One International GmbH) at 264 nm for 10 min using a BioTek Synergy H1 microplate reader (BioTek Instruments, Inc.). Initial rates of hydrolysis were plotted against the log of the boric acid and glycerol concentration and the $IC_{50}$ was calculated using the nonlinear regression function of GraphPad Prism 9 (GraphPad Software, LLC). All kinetic experiments were performed in triplicates with three repeats.

## Reporting summary

Further information on research design is available in the Nature Portfolio Reporting Summary linked to this article.

## Data availability

The structural data obtained in this study are available in the Protein Data Bank archive (https://www.rcsb.org/) under accession codes: 8pc9, 8pca, 8pcb, 8pcc, 8pcd, 8pce, 8pcf, 8pcg, 8pci, 8pcj, 8pck, 8pcl, 8pcm, 8pcn, 8pco, 8pcp, 8pcq, 8pcr, 8pcs, 8pct, 8pcu, 8pcv. Further details are available in Supplementary Tables 1 and 2. All other data obtained in this study are available from the corresponding authors upon request.

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

## Acknowledgements

We sincerely thank the staff of beamline P11 at Petra III (DESY, Hamburg, Germany) for the help with the setup of the TapeDrive and their help with data collection. We would like to thank Eva Crosas and Sofiane Saouane at this point. We acknowledge financial support from the Joachim-Herz-Stiftung Hamburg (project Infecto-Physics, given to A.P., M.P., M.A., H.R. and C.B.), the Cluster of Excellence "Advanced Imaging of Matter" of the Deutsche Forschungsgemeinschaft (DFG) – EXC 2056 – project ID 390715994, and BMBF via projects 05K19GU4 and 05K20GUB. We acknowledge financial support from the Open Access Publication Fund of Universität Hamburg.

## Author contributions

A.P., M.P., H.R. and C.B. designed the research. A.P. and N.W. purified and crystallized CTX-M-14. A.P. performed inhibition assays. D.O., A.H. and J.M. designed the current TapeDrive setup. J.M. produced the TapeDrive Nozzle based on initial design by D.O. J.H., G.P. and J.M. adapted the controls and devices of the TapeDrive to the crystallography station of the beamline P11. A.P., A.H. and D.O. performed serial crystallography data collection. M.G. and O.Y. processed serial crystallography data. D.O. wrote the script to automatically extract statistics for the datasets. A.P. and M.P. refined crystallographic protein models and A.P., M.P., and W.H. analyzed the time-resolved protein structures. H.C., M.A., H.R. and C.B. provided resources and edited the manuscript. A.P., M.P. and W.H. wrote the manuscript with input by all authors. All authors reviewed the manuscript.

## Funding

## Competing interests

The authors declare no competing interests.
