## [Peer Review File · Communications Chemistry]

This manuscript has been previously reviewed at another Nature Portfolio journal. This document only contains reviewer comments and rebuttal letters for versions considered at *Communications Chemistry*.REVIEWERS' COMMENTS:

Reviewer #1 (Remarks to the Author):

The revised manuscript has addressed the comments from the previous reviews, including the limitations of the experimental conditions and additional discussions of the reaction with glycerol. As the reviewers agreed, the study was technically sound although there were concerns about the significance related to understanding beta-lactamase enzyme mechanism and inhibitor design. Despite the limitations, the results do offer unique insights into the reactions inside the active site of an important class of enzymes, using state-of-the-art time-resolved X-ray crystallography. The discussion is also improved. One minor comment on the authors' responses to the biological relevance of the anion binding pocket. The reviewer's original comment concerns the biological relevance of the sulfate in this pocket as it is a result of the relatively high concentration of sulfate in the crystallization buffer. The anion binding pocket itself is biologically relevant since it binds to the carboxylate group of beta-lactams. But the sulfate in the crystal structure may not be biologically relevant. There appeared to be some confusion about this in the response letter. However, the discussion in the manuscript itself seemed appropriate.

Author responses

Reviewer #1 (Remarks to the Author):

The revised manuscript has addressed the comments from the previous reviews, including the limitations of the experimental conditions and additional discussions of the reaction with glycerol. As the reviewers agreed, the study was technically sound although there were concerns about the significance related to understanding beta-lactamase enzyme mechanism and inhibitor design. Despite the limitations, the results do offer unique insights into the reactions inside the active site of an important class of enzymes, using state-of-the-art time-resolved X-ray crystallography. The discussion is also improved. One minor comment on the authors' responses to the biological relevance of the anion binding pocket. The reviewer's original comment concerns the biological relevance of the sulfate in this pocket as it is a result of the relatively high concentration of sulfate in the crystallization buffer. The anion binding pocket itself is biologically relevant since it binds to the carboxylate group of beta-lactams. But the sulfate in the crystal structure may not be biologically relevant. There appeared to be some confusion about this in the response letter. However, the discussion in the manuscript itself seemed appropriate.

Response to Reviewer #1:

We appreciate the reviewer's very valuable comments and responses to our manuscript. We are also grateful for the clarification of the discussion on the anion binding site and are pleased that the discussion in the manuscript itself is appropriate.